# Vulnerability of Buildings to Meteorological Hazards: A Web-Based Application Using an Indicator-Based Approach

Maria Papathoma-Köhle [1,*], Ahmadreza Ghazanfari [2], Roland Mariacher [3], Werner Huber [3], Timo Lücksmann [4] and Sven Fuchs [1]

1 Institute for Mountain Risk Engineering, University of Natural Resources and Life Sciences, 1190 Vienna, Austria; sven.fuchs@boku.ac.at
2 Institute for Green Civil Engineering, University of Natural Resources and Life Sciences, 1190 Vienna, Austria; ahmadreza.ghazanfari@boku.ac.at
3 Moya Media OG, 5020 Salzburg, Austria; mariacher@moya-media.at (R.M.); huber@moya-media.at (W.H.)
4 Austrian Road Safety Board (Kuratorium für Verkehrssicherheit), 1100 Vienna, Austria; timo.luecksmann@kfv.at
* Correspondence: maria.papathoma-koehle@boku.ac.at

**Abstract:** Recent events have demonstrated the devastating impact of meteorological hazards on buildings and infrastructure. The possible effects of climate change on their frequency and intensity but also the rise in the value of assets may increase future risks significantly. It is crucial, therefore, for decision-makers to analyze these risks, focusing on the vulnerability of the built environment to reduce future consequences and the associated costs. However, limited studies focus on the vulnerability of buildings to meteorological hazards. The aim of the present paper is to introduce an indicator-based vulnerability assessment approach for buildings subject to three meteorological hazards (windstorms, heavy rainfall, and hail). The selection of vulnerability indicators (e.g., material, roof shape, etc.) was based on a thorough literature review. The results of an expert survey were analyzed using M-MACBETH software, and the Analytic Hierarchy Process (AHP) was used to weigh each indicator according to the expert opinions and to aggregate them into an index. A web-based application was developed that gives homeowners and other end-users the opportunity to assess the vulnerability of specific buildings by indicating the municipality, the building type, and other building characteristics. The web-application is publicly available and free of charge. The resulting index is a valuable tool for decision-makers, homeowners, authorities, and insurance companies. However, the availability of empirical damage data from real events could contribute to enhancing the performance of the presented approach.

**Keywords:** vulnerability; meteorological hazards; indicators

## 1. Introduction

Meteorological hazards including strong winds, heavy rainfall, and hail can cause casualties and damage to crops, infrastructure, and buildings. Although the impact on buildings caused by meteorological hazards in Europe is not so intense in comparison to other parts of the world (e.g., hurricanes in the Caribbean and the USA), recent events (e.g., hailstorm in Austria 2021, windstorm in Corsica 2022) have clearly shown that this impact is not insignificant. While it is not clear if and how much climate change will affect the magnitude and frequency of these hazards in the future, to reduce the associated risk scientists and practitioners need to understand and analyze the physical vulnerability of the built environment [1]. It is important to emphasize that the present paper focuses on the vulnerability of the elements at risk (in this case buildings) and not the hazard itself, the probability of occurrence, or future projections of its magnitude, frequency, and extension due to climate change. The focus is on the physical impact of three meteorological hazards (windstorms, heavy rain, and hail) on buildings. As far as

vulnerability assessment is concerned, recent reviews have emphasized that there is no universal method for analyzing, assessing, or quantifying physical vulnerability to the built environment [2]. Approaches include the development of matrices combining the intensity of a process to expected damages, vulnerability curves assessing the monetary degree of loss under different intensities, and vulnerability indicators expressing the characteristics of buildings that make them vulnerable to the impact of a natural hazard [2]. As far as meteorological hazards are concerned, some efforts for the vulnerability of various elements at risk can be found in the literature. Regarding hail, studies focusing on the vulnerability of crops [3] or solar panels [4] are relatively common. Nevertheless, a limited number of studies focusing on buildings [5] is also present. Vulnerability assessments of buildings subject to strong winds are more common for countries that experience hurricanes [6,7] and sometimes in combination with vulnerability to storm surges [8]. However, concerning heavy rainfall, most studies focus on the vulnerability to rainfall-induced hazards such as floods, landslides, and debris flows [9].

In the present paper, we adapt a well-established vulnerability assessment method [10] for meteorological hazards which had originally been developed for tsunamis and was later applied to other hazards. The selection of indicators (characteristics of the buildings) based on the impact that these three hazards have on the building shell is described in the following paragraphs. The weighting of the indicators and their aggregation in an index is then demonstrated. Following the development of three physical vulnerability indices (PVI) for buildings (one for each hazards type), the web application ([www.vulni.at](www.vulni.at), accessed on 7 March 2023) based on the described methodology is presented, which may be used by different users for risk communication and awareness-raising of affected homeowners and planning of local adaptation measures [11]. The novelty of the study lies in the fact that indicator-based approaches for the assessment of physical vulnerability to meteorological hazards are rare. The study takes a closer look at building characteristics that affect their vulnerability to meteorological hazards such as windstorms, hail, and heavy rain and engages experts in order to define their relative importance.

## 2. Materials and Methods

The concept of the method applied in this paper was originally designed for buildings subject to tsunami hazards in coastal areas [12,13]; however, it was later adapted for dynamic flooding in mountain areas [14]. Recently, the method was modified to assess the vulnerability of buildings to wildfires [15]. The approach is based on the selection of vulnerability indicators that are related to the characteristics of a building and its surroundings, their weighting, and finally their aggregation in a physical vulnerability index (PVI). The weighting of the indicators can be the result of statistical analysis of empirical data [16]. If these are not available, expert judgment and Analytic Hierarchy Process (AHP) can be also used [10]. The AHP is a method that is based on pairwise comparison decision criteria to derive weights of factors.

In the present study, a new set of indicators has been selected based on building characteristics that influence vulnerability to three meteorological hazards (windstorms, heavy rainfall, and hail). The indicators were weighted using the AHP and were finally aggregated to an index that can be assigned to each building separately. The workflow is shown in Figure 1 and the steps of the method are described in the following paragraphs. The impact of each hazard type on the building envelope and the indicators that are considered in this study are presented to justify the selection of the indicators before the detailed description of the index development.

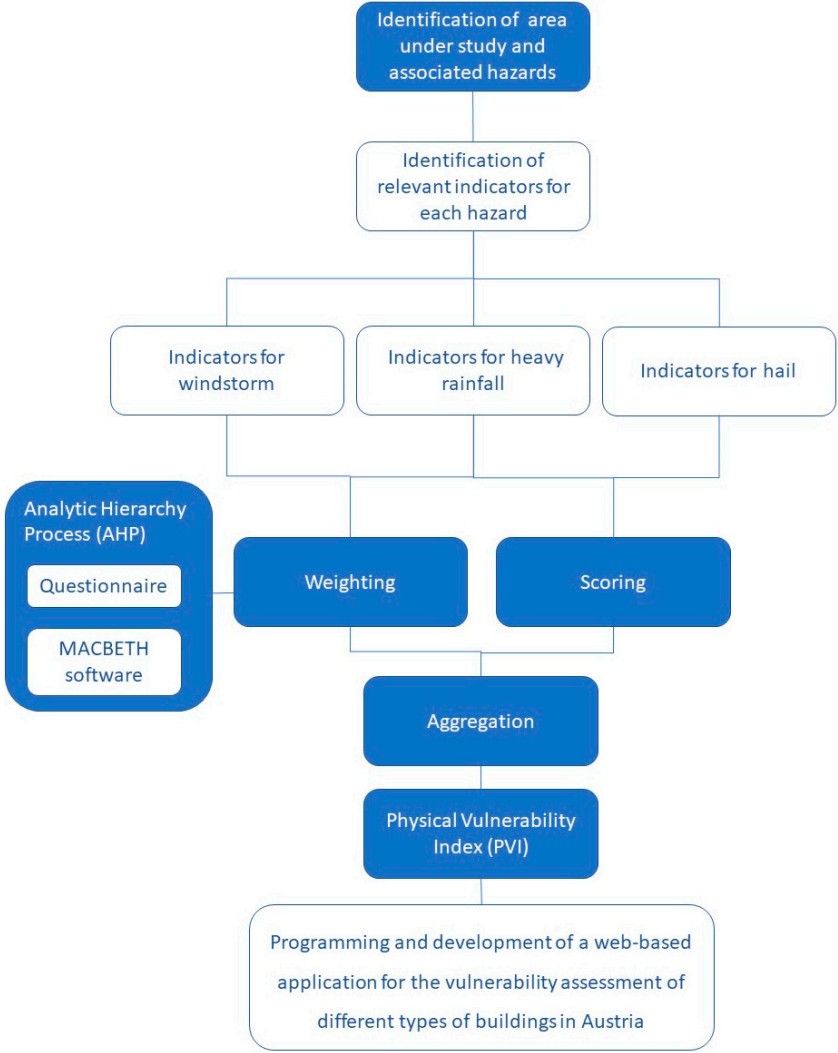

**Figure 1.** The workflow of the study.

## 2.1. The Impact of Meteorological Hazards on Buildings

The present study focuses on three meteorological hazards: windstorms, heavy rainfall, and hail. To decide which indicators are going to be used in the study, a closer look at the impact that these processes have on buildings is necessary. In the following paragraphs, the impact of the three hazardous processes on buildings is described.

### 2.1.1. Impact of Windstorms on Buildings

Windstorms are the result of the combination of opposing forces created by the development of low atmospheric pressure surrounded by a system of high atmospheric pressure [17]. A windstorm may or may not be accompanied by rain and the wind speed may exceed 55 km per hour [18]. Although it is not clear how climate change will affect the frequency and magnitude of windstorms in different parts of Europe, the overall risk may not change; however, given the value of assets in hazardous areas, it remains considerably high [19,20].

Windstorms can destroy roofs, damage walls, and break window glazing through direct contact with the wind or through broken branches, falling trees, or debris that may be carried by the wind [21]. According to Weller et al. [21], most of the damage following the impact of strong wind is recorded on roofs, sun protection systems, and facades. Weller et al. [21] also point out the importance of the roof shape, slope angle, and presence of overhangs in the resulting damage on a building, and they propose the use of storm calms for roof tiles and retrofitting of pitched roofs.

### 2.1.2. The Impact of Heavy Rain on Buildings

In the literature, definitions of heavy rainfall vary and depend on the geographical location. In Austria for example, the 98th percentile of the precipitation distribution has been chosen as a criterion to define heavy precipitation [22]. The impact of heavy rain on buildings has not been described in detail in the literature so far. Research focuses mainly on the impact of rainfall-induced hazards on buildings including flooding, landslides, and debris flows. Nevertheless, Weller et al. [23] describe the damage pattern of buildings in Germany following extensive rainfall events. They claim that the damages are mainly due to water or moisture loads on the exterior of the building. In more detail, they outline some typical impacts, including damage of the roof terraces and green flat roofs, damage to the connection points between balconies or the pitched roofs of the main buildings, and damage to the basement and exterior basement walls, as well as in underground garages. Finally, damage has also been recorded to waterproofing and its connection to the main building. Deficiencies in drainage systems and gutters have also been observed.

### 2.1.3. The Impact of Hail on Buildings

According to the WMO [24], hail is "precipitation in the form of particles of ice (hailstones)". Despite the frequent occurrence of the process, there is still a strong need to systematically collect information on hail damage [25]. The degree of damage to buildings clearly depends on the particle size. According to Munich Re [26], damage to greenhouses may occur with a hail diameter of 2 to 3 cm, whereas light roofs may be damaged by hail with a diameter of 3 to 4 cm. Windows may break after the impact of larger hail particles (diameter of 4 to 6 cm). Significant building damage, including damage to facades and window frames, has been recorded for hail as large as a tennis ball (6 to 8 cm diameter). Finally, hail with a diameter that exceeds 11 cm may be responsible for damage to the building structure. It is therefore clear that analysis of the physical vulnerability of buildings to hail has to include information regarding the roofs, windows, and building material.

### 2.2. The Selection of Indicators

### 2.2.1. Indicators for Windstorms

A thorough literature review of detailed impacts of windstorms on buildings led to the selection of the indicators demonstrated in Table 1. In more detail, the following indicators are considered:

*Roof shape*: There are different types of roofs according to their shape and each one of them reacts differently to wind force. According to Keote et al. [27], buildings with a pyramidal roof experience a lower uplift force in comparison with buildings with a gable or a hipped roof. However, wind characteristics (wind direction and angle) [28] as well as the slope of the roof [29] also play a role. A dormer window can be damaged dramatically by a windstorm if it is not protected by a shutter or with a good glazing type [30].

*Roof material*: As far as the roof material is concerned, apparently roofs made of asphalt shingles are the least resistant to wind force, whereas metal [31,32] and concrete roofs [33] perform better than clay or slate tiles.

*Roof slope*: Many studies [34,35] investigating the behavior of roofs with different declination to wind suggest that the steeper the roof slope, the lower the vulnerability of the structure to the impact of wind, setting different thresholds ranging from 20 to 45 degrees. Flat roofs are more likely to resist damage due to high wind speed, although they can still experience damage, especially along the corners and perimeter edges where wind pressure can cause uplift by the presence of eaves [21].

*Length of overhang*: A large roof overhang can cause further damage to the entire roof due to uplift when exposed to wind loads [36].

*Number of floors*: Several studies have shown that the height of the building may increase the wind pressure not only on the building itself but also on surrounding buildings [7,37,38].

*Window glazing*: The type of window glazing (single, double, or triple) plays a significant role in the overall vulnerability of the building since a broken window may increase damage in other parts of the building and its interior [7,39,40].

*Presence and type of shutter*: Based on several research studies, shutters are among the most effective mitigation/adaptation measures [41]; therefore, their presence and their type should be included in the vulnerability analysis.

*Engineering-based state of the building*: This indicator expresses the degree to which a building complies with the local building codes and standards. National codes may include standards associated with the connection between the roof and walls, materials used, and the foundation and lateral resistance of the building [42]. We include here three different codes related to wind according to the Austrian National Code. The codes are relate to: (a) the connection between walls and roofs, (b) the building foundation, and (c) the building lateral resistance.

*State of non-structural elements*: Non-structural elements such as roof components, antennas, chimneys, gutters, porches, photovoltaic panels, etc. are the most vulnerable features of a building to the impact of wind [43] due to their exposure and may increase the overall damage to the building as well as the related costs.

*Presence and state of the balcony*: The existence of balconies and the type of protection (shutter, glass, no protection) contribute significantly to the overall vulnerability of the building [44].

**Table 1.** Vulnerability indicators and their relevance to different meteorological hazard types.

| Indicators | Windstorm | Heavy Rainfall | Hail |
|---|:---:|:---:|:---:|
| Roof shape | ✓ | ✓ | ✓ |
| Roof material | ✓ | ✓ | ✓ |
| Roof slope | ✓ | ✓ | |
| Overhang length | ✓ | ✓ | ✓ |
| Number of floors | ✓ | | ✓ |
| Glazing type of window | ✓ | ✓ | ✓ |
| Presence and type of shutter | ✓ | | ✓ |
| Engineering-based state of the building | ✓ | | |
| State of non-structural elements | ✓ | | ✓ |
| Presence and state of the balcony | ✓ | ✓ | ✓ |
| Presence and state of the basement | | ✓ | |
| Presence and state of the gutter | | ✓ | |
| Presence and state of the intersections' waterproofing | | ✓ | |

2.2.2. Indicators for Heavy Rain

Heavy rain may damage different parts of the building. The following indicators were selected for consideration in this study based on the damage pattern descriptions in the literature [23] (Table 1):

*Roof slope*: The slope angle of the roof is one of the most important vulnerability indicators when it comes to heavy rainfall. In more detail, flatter roofs experience frequent damage by heavy rainfall as the water remains on the roof and the runoff is not fast, so the possibility of infiltrating into roof components increases [45].

*Presence and length of the overhang*: Foroushani et al. [46] have shown that there is a significant reduction in the amount of wind-driven rain deposition on the upper half of the facade due to roof overhangs. A survey implemented by Ge and Krpan [47] showed that a low-rise building with a typical overhang of 0.3–0.6 m and a 12-story high-rise building with a typical overhang of 0.9 m can significantly reduce the deposition of wind-driven rain on the building.

*Roof material*: Damage can take place in the interior of the building due to roof leakages and for this reason, the material of the roof has to be considered in a vulnerability analysis [48].

*Roof shape*: The shape of the roof is also of great importance and should be considered since it may or may not favor the accumulation of water on top of the building.

*Window glazing*: Infiltration through windows and doors is possible and for this reason sealing is important [49]. The window glazing, however, may be a decisive factor when the building is exposed to wind-driven rain [50].

*Presence and state of the balcony*: The balcony may be a surface that enables the accumulation of water. As such, it may favor infiltration in the interior of the building through openings (windows, balcony doors that are not sealed). However, the balcony itself may also be damaged by heavy rain. Weller et al. [23] suggest that in Germany, damage to balconies and their connections to the external walls of the building has been recorded after heavy precipitation events. In Germany, the same regulations that apply for flat roofs also apply to balconies [23].

*Presence and state of basement*: Heavy rain may be responsible for some pluvial flooding, which in this case would directly affect the basement of the building. Indirectly, apart from water damage in the basement (floors and walls), electrical installations that are located there and are significant for the functionality of the building may also be damaged [49].

*Presence and state of the gutter*: The presence, condition, and size of gutters are of major importance. Accumulated water on surfaces such as roofs and terraces has to be able to flow away [49].

*Presence and state of the intersections' waterproofing*: Intersections of the roof, such as chimneys, vents, dormers, cullies, and coverings, are particularly vulnerable to the impact of heavy rainfall [45].

### 2.2.3. Indicators for Hail

In the following paragraphs, indicators related to the impact of hail that will be considered in this study are described (Table 1).

*Material of the roof*: Brown et al. [51], using insurance claims and policy-in-force data to assess the resilience of various roofing materials to hail effects, have demonstrated that the material of a building's roof is one of the most important indicators affecting the resiliency of buildings towards hail. They also mention that the hail diameter limit for damage in asphalt shingles was found to be as low as 2.54 cm (1 in) according to laboratory impact tests, but the threshold for other items such as concrete tiles was significantly higher (5.08 cm or 2 in), which shows lower resistance of asphalt shingles in comparison to concrete tiles. It is noteworthy that metal and wooden roofs had the highest claim rates [51].

*Presence and type of shutters*: Shutters are important because they protect the windows from breaking when exposed to hail of a large diameter. This protection varies according to the material of the shutter.

*State of non-structural elements*: Non-structural elements that include roof features (antennas, chimneys, etc.) can increase the overall vulnerability of a building to the impact of hail storms [51] and consequently the damage and the associated costs.

*Roof shape*: The shape of the roof also affects the vulnerability of the building and should be considered in the index. In more detail, roofs with a slope and especially a slope perpendicular to the hail direction have demonstrated increased damages during past events.

*Presence and length of overhang*: The presence and length of overhang are included in the set of indicators due to the assumption that since the overhang protects the building from wind-driven rain it can also offer some protection to the building and its features (e.g., glass windows) when exposed to hailstorms.

*Window glazing*: The glazing (single, double, triple) of the window is a decisive factor in whether a window will break or not. Reports from a severe hail event in Germany in 2013 showed that most of the damage concentrated on roofs and windows [52].

*Presence and state of the balcony*: The presence and state of the balcony are included in the index as a sensitive protruding part of a building (see windstorm indicators).

*Number of floors*: The number of floors is also considered for buildings exposed to hail. Since hail strikes at an oblique angle (due to wind), we assume that buildings with more floors are more vulnerable when exposed to hail.

### 2.2.4. The Scoring of Indicators

The scoring of the individual indicators is a crucial step in the development of an index [2]. Following the selection of the indicators, it is clear which building characteristics will be considered in the index. In this step, the categories of these characteristics (e.g., types of roof material or roof shape) and their scoring has to be determined. Based on the literature and the expected effect of each characteristic on the vulnerability of the building, the following scores have been assigned as demonstrated in Tables 2–4. Categories that increase the vulnerability of the building (e.g., a flat roof in the case of windstorms) receive higher scoring than others that make a building less vulnerable (e.g., a hipped roof). The scoring varies for each indicator from 0.1 to 1, with 0.1 indicating that the specific characteristic (e.g., balcony with shutter) reduces the overall vulnerability of the building, whereas higher scoring, e.g., 1 (e.g., no shutter), increases its vulnerability.

**Table 2.** The scoring of the indicators for windstorms.

| | Indicators | Scoring | | | | | | | | | |
|---|---|---|---|---|---|---|---|---|---|---|---|
| 1 | **Roof shape** | | Hipped | | Gable | Dormer | | Shed | | | Flat |
| | Score | 0.1 | 0.2 | 0.3 | 0.4 | 0.5 | 0.6 | 0.7 | 0.8 | 0.9 | 1 |
| 2 | **Roof material** | | Metal sheets | | Clay and concrete tiles | Slate tiles | | Wood shingles | | | Asphalt shingles |
| | Score | 0.1 | 0.2 | 0.3 | 0.4 | 0.5 | 0.6 | 0.7 | 0.8 | 0.9 | 1 |
| 3 | **Roof slope** | | 21°–30° and >30° | | | | 11°–20° | | | | 0°–10° |
| | Score | 0.1 | 0.2 | 0.3 | 0.4 | 0.5 | 0.6 | 0.7 | 0.8 | 0.9 | 1 |
| 4 | **Overhang length** | | <20 in | | | | >20 in | | | | |
| | Score | 0.1 | 0.2 | 0.3 | 0.4 | 0.5 | 0.6 | 0.7 | 0.8 | 0.9 | 1 |
| 5 | **Number of floors** | | One floor | | | | | | Two floors | | Three floors or more |
| | Score | 0.1 | 0.2 | 0.3 | 0.4 | 0.5 | 0.6 | 0.7 | 0.8 | 0.9 | 1 |
| 6 | **Window glazing** | | Triple glazing | | Double glazing | | | | Single glazing | | |
| | Score | 0.1 | 0.2 | 0.3 | 0.4 | 0.5 | 0.6 | 0.7 | 0.8 | 0.9 | 1 |
| 7 | **Presence and type of shutter** | | Metal shutter | | Composite shutter | PVC shutter | Wooden shutter | | | | No shutter |
| | Score | 0.1 | 0.2 | 0.3 | 0.4 | 0.5 | 0.6 | 0.7 | 0.8 | 0.9 | 1 |
| 8 | **Eng.-based state** | Code-based (3/3) | Code-based (2/3) | | Code-based (1/3) | | | | | | Not code-based |
| | Score | 0.1 | 0.2 | 0.3 | 0.4 | 0.5 | 0.6 | 0.7 | 0.8 | 0.9 | 1 |
| 9 | **State of non-structural elements** | | Good | | | | Normal | | | | Bad |
| | Score | 0.1 | 0.2 | 0.3 | 0.4 | 0.5 | 0.6 | 0.7 | 0.8 | 0.9 | 1 |
| 10 | **Presence and state of balcony** | With shutter | With glass | | | With awning | | With overhang or upper balcony | | | No protection |
| | Score | 0.1 | 0.2 | 0.3 | 0.4 | 0.5 | 0.6 | 0.7 | 0.8 | 0.9 | 1 |

**Table 3.** The scoring of the indicators for heavy rain.

| # | Indicators | 0.1 | 0.2 | 0.3 | 0.4 | 0.5 | 0.6 | 0.7 | 0.8 | 0.9 | 1 |
|---|---|---|---|---|---|---|---|---|---|---|---|
| 1 | **Roof slope** | | 21°–30° and >30° | | | | 11°–20° | | | | 0°–10° |
| | Score | 0.1 | 0.2 | 0.3 | 0.4 | 0.5 | 0.6 | 0.7 | 0.8 | 0.9 | 1 |
| 2 | **Overhang length** | | >20 in | | 10–20in | | | | 5–9 in | | No overhang |
| | Score | 0.1 | 0.2 | 0.3 | 0.4 | 0.5 | 0.6 | 0.7 | 0.8 | 0.9 | 1 |
| 3 | **Roof material** | | Metal sheets | | Clay and concrete tiles | Slate tiles | | Wood shingles | | | Asphalt shingles |
| | Score | 0.1 | 0.2 | 0.3 | 0.4 | 0.5 | 0.6 | 0.7 | 0.8 | 0.9 | 1 |
| 4 | **Roof shape** | | Hipped | | Gable | Dormer | | Shed | | | Flat |
| | Score | 0.1 | 0.2 | 0.3 | 0.4 | 0.5 | 0.6 | 0.7 | 0.8 | 0.9 | 1 |
| 5 | **Window glazing** | | Triple glazing | | Double glazing | | | | Single glazing | | |
| | Score | 0.1 | 0.2 | 0.3 | 0.4 | 0.5 | 0.6 | 0.7 | 0.8 | 0.9 | 1 |
| 6 | **Presence and state of balcony** | With shutter | With glass | | | With awning | | With overhang or upper balcony | | | No protection |
| | Score | 0.1 | 0.2 | 0.3 | 0.4 | 0.5 | 0.6 | 0.7 | 0.8 | 0.9 | 1 |
| 7 | **Presence and state of basement** | | Water-proofed and protection | | | | Waterproof and no protection or vice versa | | | | Not water-proofed and no protection |
| | Score | 0.1 | 0.2 | 0.3 | 0.4 | 0.5 | 0.6 | 0.7 | 0.8 | 0.9 | 1 |
| 8 | **Presence and state of the gutter** | | Good | | | | Normal | | | | No gutter |
| | Score | 0.1 | 0.2 | 0.3 | 0.4 | 0.5 | 0.6 | 0.7 | 0.8 | 0.9 | 1 |
| 9 | **Presence and state of the intersections water proofing** | | Good | | | | Normal | | Bad | | No water proofing of inter-sections |
| | Score | 0.1 | 0.2 | 0.3 | 0.4 | 0.5 | 0.6 | 0.7 | 0.8 | 0.9 | 1 |

**Table 4.** The scoring of the indicators for hail.

| # | Indicators | 0.1 | 0.2 | 0.3 | 0.4 | 0.5 | 0.6 | 0.7 | 0.8 | 0.9 | 1 |
|---|---|---|---|---|---|---|---|---|---|---|---|
| 1 | **Roof material** | | Metal sheets | | Clay and concrete tiles | Slate tiles | | Wood shingles | | | Asphalt shingles |
| | Score | 0.1 | 0.2 | 0.3 | 0.4 | 0.5 | 0.6 | 0.7 | 0.8 | 0.9 | 1 |
| 2 | **Presence and type of shutter** | | Metal shutter | | Composite shutter | PVC shutter | Wooden shutter | | | | No shutter |
| | Score | 0.1 | 0.2 | 0.3 | 0.4 | 0.5 | 0.6 | 0.7 | 0.8 | 0.9 | 1 |
| 3 | **State of non-structural elements** | | Good | | | | Normal | | | | Bad |
| | Score | 0.1 | 0.2 | 0.3 | 0.4 | 0.5 | 0.6 | 0.7 | 0.8 | 0.9 | 1 |
| 4 | **Roof shape** | | Hipped | | Gable | Dormer | | Shed | | | Flat |
| | Score | 0.1 | 0.2 | 0.3 | 0.4 | 0.5 | 0.6 | 0.7 | 0.8 | 0.9 | 1 |
| 5 | **Overhang length** | | >20 in | | 10–20in | | | | 5–9 in | | No overhang |
| | Score | 0.1 | 0.2 | 0.3 | 0.4 | 0.5 | 0.6 | 0.7 | 0.8 | 0.9 | 1 |
| 6 | **Window glazing** | | Triple glazing | | Double glazing | | | | Single glazing | | |
| | Score | 0.1 | 0.2 | 0.3 | 0.4 | 0.5 | 0.6 | 0.7 | 0.8 | 0.9 | 1 |
| 7 | **Presence and state of balcony** | With shutter | With glass | | | With awning | | With overhang or upper balcony | | | No protection |
| | Score | 0.1 | 0.2 | 0.3 | 0.4 | 0.5 | 0.6 | 0.7 | 0.8 | 0.9 | 1 |
| 8 | **Number of floors** | | One floor | | | | | | | Two floors | | Three floors or more |
| | Score | 0.1 | 0.2 | 0.3 | 0.4 | 0.5 | 0.6 | 0.7 | 0.8 | 0.9 | 1 |

### 2.3. The AHP Method and the Weighting of Indicators

The AHP, developed by Saaty [53], is one of the most common and popular participatory approaches for the weighting of indicators. The method has many advantages, including the possibility to include qualitative and quantitative data as well as the transparency of the indicators. Nevertheless, the pairwise comparison makes its implementation time-consuming for the participating experts and computationally expensive. Moreover, results based on expert judgment depend on the experts that have contributed, as well as on their knowledge, experience, and interests [54]. The indicators were first weighted pairwise by the authors, and then the MACBETH software (Measuring Attractiveness through a Category Based Evaluation Technique) was used for the analysis of the results. MACBETH is a specially designed computer program for multi-criteria decision analysis. It was used to implement pairwise comparisons of indicators and obtain relative weights. The main difference between MACBETH and other MCDA (multi-criteria decision analysis) methods is that to generate numerical scores for the options in each criterion and to weight the criteria, MACBETH only requires qualitative judgments (e.g., weak, moderate, strong) about the difference of attractiveness between two elements at a time. The weighting approach followed at this point was the same by Dall'Osso [10,55]. The approach of Dall'Osso was designed for tsunamis. The present approach is based on the same weighting method but on a different set of indicators associated with the impact of strong wind, heavy rainfall, and hail on the buildings.

A questionnaire was developed and sent to the experts to diminish the subjectivity and bias of the given weights and to finalize the weighting of the indicators. The questionnaire consisted of two parts. In the first part, the indicators selected for each hazard were introduced and input by the experts was required. In addition, a brief explanation of each indicator and their importance was given to support the experts in the weighting of the indicators. In the second part, a blank area was provided for unstructured comments on the model itself, additional information on the recommended weights, suggestions for indicators that should be also considered, and other feedback.

The questionnaire was sent to 15 experts from various backgrounds (structural engineering, insurance industry, and natural disasters research) to determine different weights for indicators according to their knowledge of building engineering and experience with past events, respectively. Nevertheless, 8 experts (more than 50%) responded to the survey. The next step was the analysis of the questionnaires by MACBETH.

Firstly, the indicators (options) were introduced to the software and by using an evaluation matrix, the pairwise comparisons between indicators affecting the structural vulnerability of buildings for each hazard were conducted. The indicators in each row and column were contrasted with each other. When it was determined that an indicator in a row was more significant than an indicator in a column, their qualitative difference was stated in the appropriate cell. The options for the difference between the two factors' relative weights were the semantic categories. Furthermore, as imaginary references, lower and higher factors were introduced. The higher one is equally significant as the most important element (engineering-based state of the building for windstorm), but the lower one makes no contribution to the degree of structural vulnerability.

The results of the survey and the pairwise comparison (AHP) led to the final weights for each indicator that are shown in Table 5.

**Table 5.** The weights of indicators for each hazard type.

| Indicators | Windstorm | Heavy Rainfall | Hail |
|---|---|---|---|
| Roof shape | 90 | 97 | 85 |
| Roof material | 90 | 86 | 100 |
| Roof slope | 53 | 99 | |
| Overhang length | 45 | 49 | 54 |
| Number of floors | 30 | | 44 |
| Glazing type of window | 65 | 43 | 69 |
| Presence and type of shutter | 71 | | 74 |
| Engineering-based state of the building | 98 | | |
| State of non-structural elements | 78 | | 87 |
| Presence and state of the balcony | 20 | 25 | 32 |
| Presence and state of the basement | | 61 | |
| Presence and state of the gutter | | 69 | |
| Presence and state of the intersections' waterproofing | | 76 | |

## *2.4. Aggregation and Final Index*

The final step is the aggregation of the weighted indicators in a single vulnerability index that can be assigned to each building for each of the three meteorological hazard types. The aggregation follows the approach of the index presented by Kappes et al. [56] for debris flows, landslides, and floods (Figure 2).

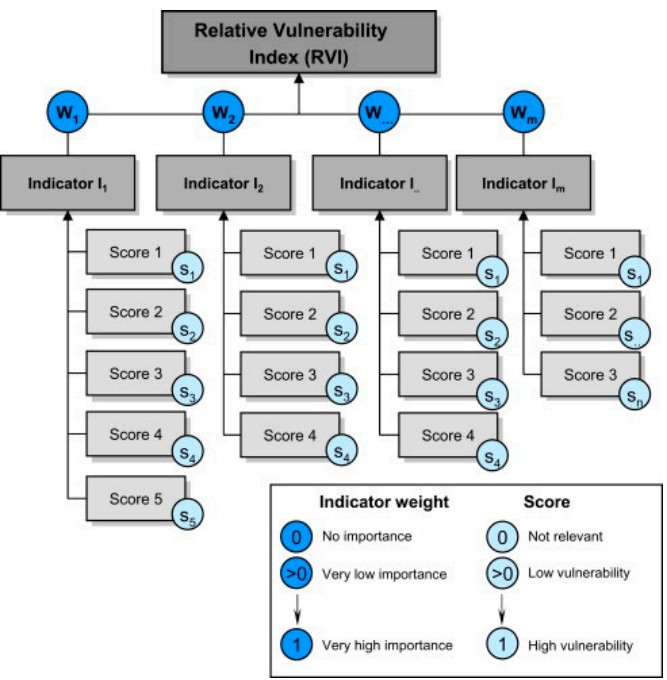

**Figure 2.** Aggregation of the indicators into an index [56].

After the weights of each indicator are defined with the AHP method, the indicators need to be aggregated into an index for each hazard according to the following equation:

$$PVI(W, HR, H) = \sum_{1}^{n} \frac{Wn}{\sum_{1}^{n} Wn} \tag{1}$$

where:

*PVI:* the physical vulnerability index;
*W:* wind hazard;
*HR:* heavy rain hazard;

*H:* hail hazard;

*n:* the number of indicators;

*Wn:* the weight assigned to indicator *n*.

The resulting index can be assigned to each building under study. The indicators can populate a GIS database and the resulting spatial pattern of the index can be visualized in a map. A more sophisticated application of the index, that of a web-based tool, is described in the following paragraphs.

## 3. Web-Based Application

The methodology described above laid the foundation for a subsequent web-based implementation. Homeowners from Austria, but also other users (local authorities, emergency services, engineers, insurance companies), can use this tool to self-assess the vulnerability of their house in the framework of their municipality. In this section, the implementation and application of the web tool are described and discussed.

### 3.1. Architecture of the Web Application

The platform was realized as a dynamic, responsive web application. Users enter information about their building and municipality in multiple steps and receive a dynamically generated report with information in the form of text, images, and diagrams. The report can be accessed via a unique ID created at the end of the process within the web application and can be shared with others via web link as well as saved locally. Due to the significant number of parameters that users need to enter to receive a meaningful report, great emphasis was placed on providing information in a gentle manner. The individual building parameters are structured in multiple models and the progress of the input is visualized through UI elements. Further information on often non-self-explanatory technical terms has been realized in the form of overlays, which can be activated by the user when needed. The backend of the application consists of a PHP application in conjunction with a relational MySQL database (version 8.0.31). Several PHP scripts handle tasks for transmitting and validating submitted data as well as making it available for visualization in the front end of the web application. Here, the challenge was to consolidate data from multiple sources and in different formats into a single database while keeping the system as performant as possible.

### 3.2. Implementation of the Vulnerability Index

In a first step, the user must choose the building type to be assessed. These building types are representative of the majority of residential buildings in Austria [57] (see Figure 3). In more detail, the building typology includes the following building types:

1. Worker house (*Arbeiterhaus*): The construction of this type of building started in the 1950s. This type of building is an old-style, single-family house common in historically industrialized areas in Austria. Moreover, the building materials used are of lower quality according to the post-WW II building practices.

2. Single-family house (*Einfamilienhaus*): A single-family house is a residential building where one family lives (one residential unit). The construction period for this building type started in the 1960s.

3. Country house (*Landhaus*): A country house is located in the countryside, and it often has a large and heavy roof and one or two floors. Furthermore, a lot of real wood is used for country houses. Other defining features are large wooden balconies and lattice windows made of wood with window shutters. Hip and gable roofs are particularly common for the roof shape, but there are numerous variations possible. In addition, the construction of this building prototype started the 1950s.

4. Row house (*Reihenhaus*): Row houses are single-family houses that are built in a closed row (terraced housing) with at least two homes that have a similar design. The side walls that are shared with the adjacent house must be double-walled and without windows, and a small garden is another feature of most row houses. Furthermore,

row houses are located in urban areas or near cities. This type of building began to be constructed in 2000.

5.  Apartment building (*Wohnblock*): These types of buildings are comprised of large multi-family and multi-story residential structures with more than eleven housing units and are mostly found in the central part of urban areas. The construction period of this type of building started in the 1980s.

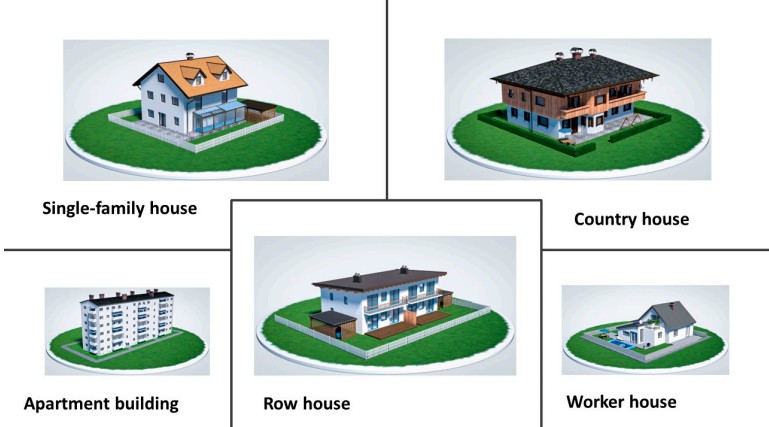

**Figure 3.** Building typology for residential buildings in Austria.

In the second step, the user must define the construction period, the number of floors, and the geographical location (95 districts plus the 23 districts of Vienna) of the building using the respective drop-down menus.

Next, the online form requires the roof type (Figure 4) and inclination, the roof material and overhang length, and the condition of sealing and tightening of connections, installations, and gutters. In the fourth step, information about the window type and glazing as well as the building openings, other outbuildings, and shutters has to be provided using drop-down menus. In the last step, information on the basement, sewage system, and the surroundings of the building is needed.

Finally, following the submission of the above information, the web tool produces a summarizing report indicating the overall building vulnerability towards different hazards and the vulnerability of specific components (see Figure 5 as an example). Additional information includes a statistical overview of the municipality.

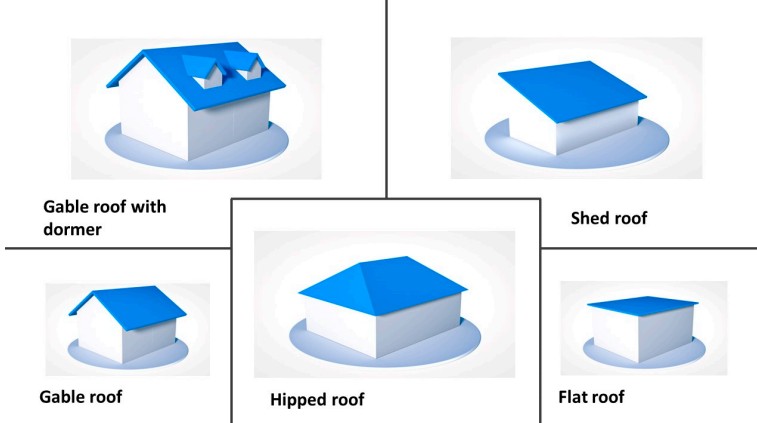

**Figure 4.** Roof typology for residential buildings in Austria.

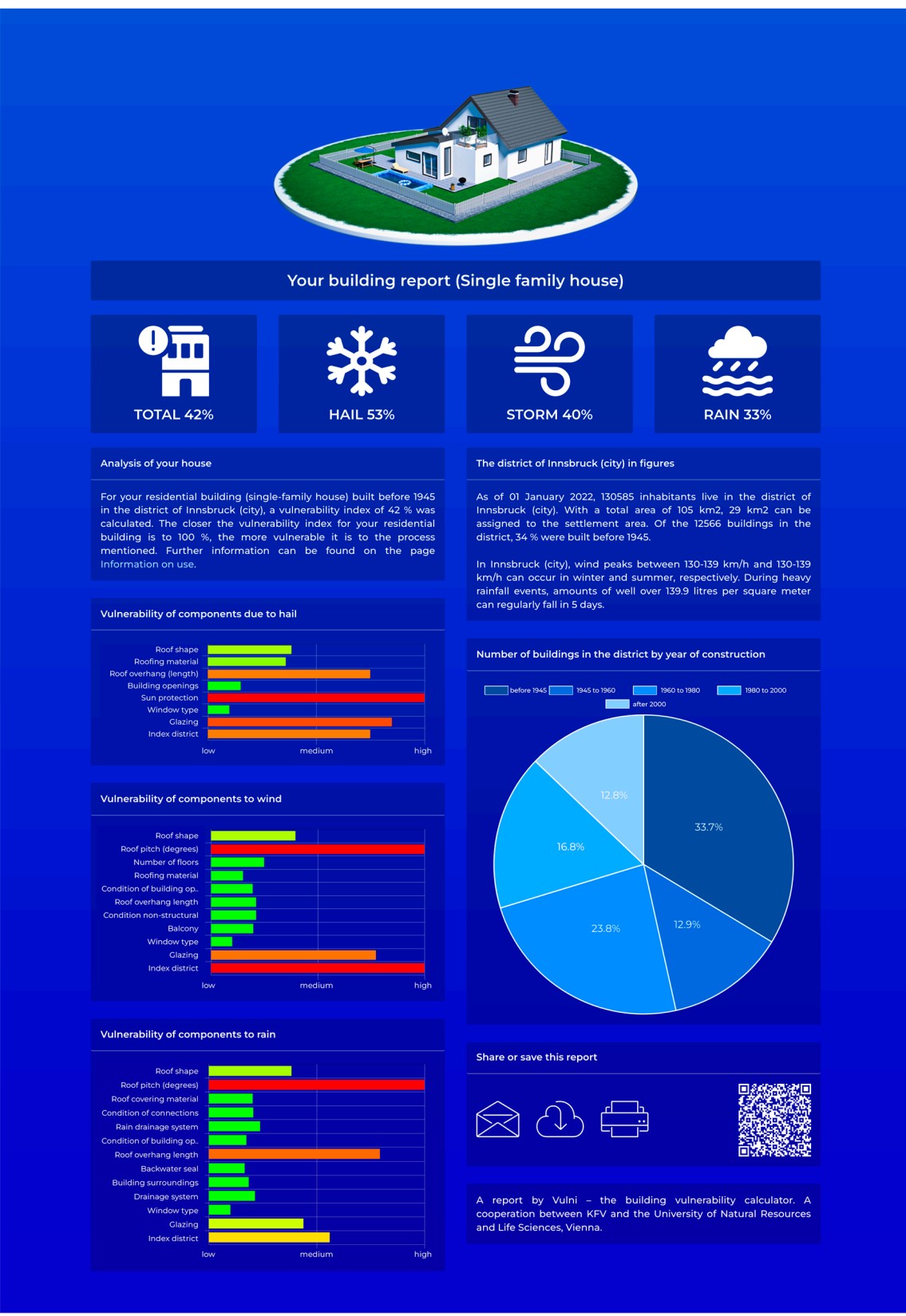

**Figure 5.** Example for a final vulnerability profile provided by the web tool.

The report can easily be shared or downloaded using provided links and a QR code. The tool is free of charge and publicly available (www.vulni.at, accessed on 7 March 2023).

Future adaptations of the website have already been considered in the concept phase, so changes can be made with very little effort.

## 4. Limitations and Further Developments of the Study

The study presented herein describes a well-established vulnerability assessment method modified for three meteorological hazards. The approach uses a new set of indicators for each hazard based on available literature review. These indicators are weighted by using expert judgment and the AHP method and then are aggregated in an index. It is important to note that there were some limitations in the study. The most important limitation is the lack of indicators related to the phenomenon itself and its intensity. We assume that the building will be damaged by a potential hazard, e.g., a hail storm, but we do not define the intensity of this storm or its characteristics (e.g., hail grain size). This limitation should inspire a future development of the study where the intensity can also be considered in the index. This has been achieved in the case of tsunamis [10] and dynamic flooding [14] by including the intensity of the process, e.g., the water vulnerability (part of the building submerged in the water). Moreover, the spatial pattern of the phenomenon under study could also be included and overlayed with the vulnerability of buildings belonging to a settlement. This could guide emergency management decisions and local adaptation measures. Another limitation is the scoring of the indicators. The scoring as presented herein is based on a review of the existing scientific literature, which is limited. The scoring should be revised in the future and should be seen in combination with the characteristics of the potential meteorological hazard. Finally, future projections of the meteorological hazards discussed herein could also be included and combined with the vulnerability index to identify future catastrophic scenarios and potential negative consequences. Another limitation which is common in studies including expert judgment is the choice, background, number, response rate, and subjectivity of the experts involved. A higher number of experts, the involvement of international experts, or the involvement of experts belonging to the same group (e.g., engineers) could also be a future development of the study.

## 5. Conclusions

The use of indicators for vulnerability assessments of buildings subject to three meteorological hazards and the implementation in a web tool has been presented and discussed in this paper. A thorough literature review led to a set of vulnerability indicators for each building that were weighted using expert judgment and AHP. The vulnerability indicators were aggregated in a single index. The methodology presented here is of great value to a variety of end-users. Homeowners can assess the vulnerability of the building and reduce it accordingly. Local authorities can have a detailed picture of the physical vulnerability and its spatial variation within a settlement and use this information as a basis for decision-making. Insurance companies, engineers, and scientists, but also emergency services, can use the generated information for, e.g., setting premiums, investigating the interaction between structures and natural processes, and optimizing the design of buildings. Indicator-based vulnerability analysis for buildings subject to meteorological hazards is, however, still in its infancy. Further research towards a validation of the existing methodology following new events is required. Furthermore, laboratory experiments testing different roof materials, for example, could be used to refine the indicator selection. Statistical methods could be used for the weighting of indicators based on empirical damage data. Weighting based on expert judgment can be improved by including more experts from different countries and disciplines. Additionally, more indicators should be considered, including indicators associated with the immediate surroundings of the buildings, such as shielding from dense vegetation or other buildings and the existence of movable objects in the immediate vicinity. Finally, the interaction between these three hazards on the impact on buildings should also be investigated since these three hazards often occur at the same time.

This method involves several uncertainties and limitations, such as high data requirements, time-consuming data collection, and uncertainties associated with the reaction of

materials and building components. Nevertheless, the indicator-based approach presented in this paper provides an overview of the vulnerability of buildings and their vulnerable components at a local scale, giving the opportunity to end-users to target specific buildings or weak points of these buildings and prioritize their actions and their resources for vulnerability and risk reduction.

**Author Contributions:** Conceptualization, T.L., M.P.-K., A.G. and S.F.; methodology, M.P.-K. and A.G.; software, W.H. and R.M.; data curation, S.F., W.H. and R.M.; writing—original draft preparation, M.P.-K.; writing—review and editing, S.F., W.H. and R.M.; visualization, W.H. and R.M.; supervision, S.F.; project administration, T.L.; funding acquisition, S.F. and T.L. All authors have read and agreed to the published version of the manuscript.

**Funding:** This research received funding from the Austrian Road Safety Board (Kuratorium für Verkehrssicherheit, KFV), Austria. Please note that continuous hosting and maintenance of the website is provided by KFV.

**Institutional Review Board Statement:** Not applicable.

**Informed Consent Statement:** Not applicable.

**Data Availability Statement:** No new data were created.

**Conflicts of Interest:** The authors declare no conflict of interest.

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
