# Peer review of "Vulnerability of Buildings to Meteorological Hazards: A Web-Based Application Using an Indicator-Based Approach"

_applsci, doi:10.3390/app13106253_

Round 1

Reviewer 1 Report

General Comments

The work presented concerns construction of a vulnerability index for buildings that can be built based on a web application.
The main motivation for constructing such an index has two reasons: the damage caused in various parts of the world by extreme weather events, the available forecasts projecting weather parameters in climate change in dependence on possible global warming scenarios (IPCC 6th Assessment Report, 2019).
The constructed indices are based on the extrapolation of damages of different types of weather factors and the combination of these factors. This implies that extreme events for one parameter are reflected in the nonlinear variability of the parameter combination of the three risk parameters. The statistics of these parameters reflect the variability of winds, rain and meteors in general, including hail. Since these parameters are not linearly correlated with each other and with increasing temperatures throughout Europe and thus also in Austria, the statistics and related risks are difficult to predict with such functions.
Nevertheless, in quite general terms, the calculated index can give a broad measure of the correlated risks for the specific buildings considered and for the nation considered.

Specific comments:

l. 99-101 Windstorms are defined by dictionaries as a storm consisting as violent winds or violent gusts but little or no rain. The intended meaning in this paper is strong winds resulting from thunderstorms or other weather phenomena that can result from many processes whose consequence is strong winds. Destructive winds is what is being talked about.

In fact, the IPCC Sixth Report is cited as the source of future projections of wind intensity in general, resulting from the increase in extreme events where wind is the considered risk factor.  The conclusion of the cited study, is that the analysis of wind data in projections is limited by the spatio-temporal resolution of the simulations that project the data into the future. Thus, this resolution is unable to capture intense events in the local resolution needed to measure these parameters. This does not mean that there are no models that are able to resolve the measurement of winds downscaled by such simulations. An accurate index must start with such data.  

l. 333 If the indicators being combined do not contain or only partially contain the variability associated with the phenomena being discussed, the index developed, even having counted with a reasonable weight the error associated with a certain risk, contains no indication of the possible risk for, e.g., the change in precipitation intensity in the near future. The nonlinearity of the meteorological processes discussed does not permit the use of a linear combination such as that used for risk counting. Such a combination will probably be fine for a good portion of the risks considered, but not for the extreme risks that will occur in the future.  

Author Response

Dear Reviewer 1,

Thank you very much for your constructive comments. Please find attached our response to each one of them.

Many thanks

Kind regards

The authors

Reviewer 2 Report

The purpose of this work was to develop an indicator-based vulnerability assessment methodology for structures susceptible to three meteorological risks (windstorms, heavy rainfall, and hail). The paper's methods lay the groundwork for a subsequent web-based implementation. 

This paper presents and discusses the use of indicators for vulnerability assessments of structures vulnerable to three meteorological hazards and the integration in a web platform. What are the newest of the approach, potential improvements to the methodology, and potential future study of the paper?  
The use of MACBET to compute weight is not described in depth; therefore, it should be explained how weight is calculated.

Author Response

Dear Reviewer 2,

Many thanks for your comments. Please find our response in the attached file

Kind regards

The authors

Reviewer 3 Report

For this reviewer it should be stated how much severer the analysed meteorological condition are in compare with historical data. Not only of the frequency but intensity. Without this fact taking into the consideration this paper has no any meaning it could be stated that all the building engineering knowledge stated in standards and best practices doesn't matter.

The novelty is hardly emphasized.

How authors define the described hazards in this study? And how these hazards are different from hazards presented in the historical observations?

This reviewer suggests consulting this work with a building engineer.

This paper suffers deficiency in the methodology section and absolutely lack of the results and discussion section.

Have authors conducted any statistical analysis of data from the questionnaire?

Additional comments in the attached version.

Author Response

Dear Reviewer 3,

Thank you very much for your constructive comments. please find our response in the attached file

Kind regards

The authors
